# Excitonic density wave and spin-valley superfluid in bilayer transition metal dichalcogenide

Zhen Bi [1,2 ✉] & Liang Fu[1 ✉]

Artificial moiré superlattices in $2d$ van der Waals heterostructures are a new venue for realizing and controlling correlated electronic phenomena. Recently, twisted bilayer $WSe_2$ emerged as a new robust moiré system hosting a correlated insulator at moiré half-filling over a range of twist angle. In this work, we present a theory of this insulating state as an excitonic density wave due to intervalley electron–hole pairing. We show that exciton condensation is strongly enhanced by a van Hove singularity near the Fermi level. Our theory explains the remarkable sensitivity of the insulating gap to the vertical electric field. In contrast, the gap is weakly reduced by a perpendicular magnetic field, with quadratic dependence at low field. The different responses to electric and magnetic field can be understood in terms of pair-breaking versus non-pair-breaking effects in a BCS analog of the system. We further predict superfluid spin transport in this electrical insulator, which can be detected by optical spin injection and spatial-temporal imaging.

[1] Department of Physics, Massachusetts Institute of Technology, Cambridge, MA, USA. [2] Department of Physics, the Pennsylvania State University, University Park, PA, USA. ✉email: zjb5184@psu.edu; liangfu@mit.edu

A new era of band engineering and novel electronic phase design is promised by the discovery of moiré superlattices in 2d van der Waals heterostructures. Recent experiments on graphene-based moiré systems[1–13] unfolded a brand new world of strongly interacting electron phases, including correlated insulating states, unconventional superconductivity as well as quantum anomalous hall states. Moiré superlattices based on transition metal dichalcogenides (TMD)[14–19] lately emerge as a new host of novel electronic phases of matter. Unlike graphene, the semiconducting TMDs possess a large bandgap and large effective electron mass, as well as strong spin–orbit coupling. Due to these features, narrow moiré bands appear generically in heterobilayer[15] or twisted homobilayer TMDs[14] without fine-tuning, which provides a fertile ground for correlation physics. In addition, the spin-valley locking in TMDs[20–22] offers unique opportunities to probe and manipulate electron spins with optical methods[17,23].

In a very recent experiment on WSe$_2$ homobilayers[18] with twist angles $\theta \in [4°, 5.1°]$, a striking correlated insulating dome was found at low temperature within a narrow range of the vertical electric field, when the relevant moiré band is half-filled. The activation energy gap of the insulating state is ~3 meV, which is remarkably large among most moiré systems[1–3,8–12]. However, the insulating gap is smaller by an order of magnitude than either the miniband width (about $60-100$ meV for $\theta = 4-5°$) or the characteristic Coulomb interaction energy $e^2/\epsilon L$ (~30 meV for a dielectric constant $\epsilon = 10$ at $4°$). The smallness of the insulating gap compared to the interaction energy and bandwidth and its sensitivity to the displacement field speak against the scenario of a Mott insulator and call for a new theoretical understanding.

In this work, we predict that the insulator in twisted TMD at half-filling is an excitonic density wave formed by the pairing of electrons and holes in minibands at different valleys. This electron–hole pairing is strongly enhanced by a van Hove singularity[24–31] (VHS) near the Fermi level. We show that the detuning of the displacement field has a pair-breaking effect on the excitonic insulator, while the Zeeman coupling to an out-of-plane magnetic field is non-pair-breaking at leading order. We introduce a low-energy theory for twisted TMD and calculate the phase diagram as a function of the displacement field, temperature, and magnetic field, finding good agreement with the experiment[18].

Remarkably, owning to spin-valley locking in TMD[32,33], our excitonic insulator is also a spin superfluid and thus enables coherent spin transportation over long distances. The possibility of spin supercurrent was initially theorized in magnetic insulators with easy-plane anistropy[34–38]. Its signatures have been reported in recent electrical measurements on quantum Hall state in graphene and antiferromagnetic insulator $Cr_2O_3$[39–41], where spin Hall effect is used to generate and detect a nonequilibrium spin accumulation. A key advantage of 2d TMDs is that the spin polarization can be easily generated and detected by purely optical means. It has been shown that a circularly polarized light can efficiently generate spin polarizations in TMDs due to the spin-valley locking and valley-selective coupling to chiral photons[20–22]. The local spin polarization can be read out by measuring the difference in reflectance of right- and left-circularly polarized lights, i.e., the circular dichroism spectroscopy[17,23]. Therefore, twisted TMDs provide an ideal platform for studying spin transport with a fully optical setup. We propose an all-optical setup for spin injection and spatial–temporal imaging of spin transport.

## Results
**Moiré band structure**. We consider the electronic band structure of twisted homobilayer WSe$_2$. At a small twist angle, the two sets of moiré bands in bilayer TMD originating from $K$ and $K'$ valleys are decoupled at the single-particle level and treated separately hereafter. From the outset, it is important to distinguish moiré systems generated by slightly twisting AA and AB stacking bilayer TMDs—which differ by a 180° rotation of the top layer. The two cases have very different moiré band structures due to spin-valley locking in TMD. For the AA case, the $K$ valleys on two layers with the same spin polarization are nearly aligned, so that interlayer tunneling is allowed and creates layer-hybridized moiré bands, as shown in Fig. 1a. In contrast, for AB, the $K$ valley on one layer is nearly aligned with the $K'$ valley on the other layer with the opposite spin polarization, so that interlayer tunneling is forbidden and the resulting moiré bands have additional spin/layer degeneracy[14,42]. Throughout this work, we consider the moiré system from twisting AA stacking bilayer TMD, where the full filling for the topmost moiré bands corresponds to two electrons per supercell.

We calculate the moiré band structures using the continuum model from ref. [14] with parameters extracted from first principle calculations (see the section "Moiré band structure" in "Methods"). Remarkably, we find that the moiré band dispersion is highly tunable by the displacement field, $D$. For a certain range of displacement field, our calculation (Fig. 1b, c) shows that the Fermi level at half-filling is close to a van Hove singularity (VHS) in the density of states, resulting from saddle points near the corners of the mini-Brillouin zone, $\mathbf{K_M}$ and $\mathbf{K'_M}$. The proximity to VHS is supported by the observed sign change of Hall coefficient near half-filling[18]. In the relevant parameter regime, the moiré band has a nontrivial Chern number. Details of the topological properties of the moiré band are discussed in the section "Band topology and Berry curvature". While playing an important role in the flat-band scenario in graphene-based moiré systems[43–45], the topological property does not appear to play a significant role in the weak coupling scenario considered here.

Since a diverging DOS near the Fermi level enhances correlation effects, the detuning of VHS by the displacement field is expected to affect the metal–insulator transition. Motivated by this consideration, we now develop a low-energy theory by expanding the moiré band from valley $K$ ($K'$) around $\mathbf{K_M}$ ($\mathbf{K'_M}$). Based on the lattice symmetries, the Taylor expansion up to the third order in momentum takes the general form

$$\varepsilon_\pm(\mathbf{k}) = \alpha \mathbf{k}^2 \pm \xi(\mathbf{k}), \quad \xi(\mathbf{k}) = k_y^3 - 3k_y k_x^2 \qquad (1)$$

where $\pm$ is the valley index related by time-reversal symmetry $\mathcal{T}: c_{\mathbf{k},+} \rightarrow c_{-\mathbf{k},-}, c_{\mathbf{k},-} \rightarrow -c_{-\mathbf{k},+}$.

The coefficient $\alpha$ depends on the displacement field $D$ (see "Methods" for more details). It is useful to first consider a critical displacement field $D_c$ where $\alpha = 0$. Then, $\mathbf{k} = 0$ becomes a monkey saddle point where three energy contour lines interact[29,46], resulting in a high-order van Hove singularities (hVHS) with a power-law-divergent density of states[27–29,46]

$$\rho_{\alpha=0}(E) \sim 1/|E|^{1/3}, \qquad (2)$$

as shown in Fig. 1c.

**Intervalley exciton order**. At low carrier density, the dominant interaction is Coulomb repulsion within the same valley and between two valleys: $H_{int} = \sum_{i,j} \int d\mathbf{r} d\mathbf{r}' V_{i,j}(\mathbf{r} - \mathbf{r}') n_{i,\mathbf{r}} n_{j,\mathbf{r}'}$, where $i, j = \pm$ is valley index and $n_{i,\mathbf{r}} = c_{i,\mathbf{r}}^\dagger c_{i,\mathbf{r}}$ is density operator of a given valley. The interacting Hamiltonian is reminiscent of bilayer quantum Hall system[47], where the layer degree of freedom plays the role of the valley. At half-filling of each layer, Coulomb repulsion leads to interlayer exciton condensation. However, unlike Landau levels, here the moiré band in twisted bilayer

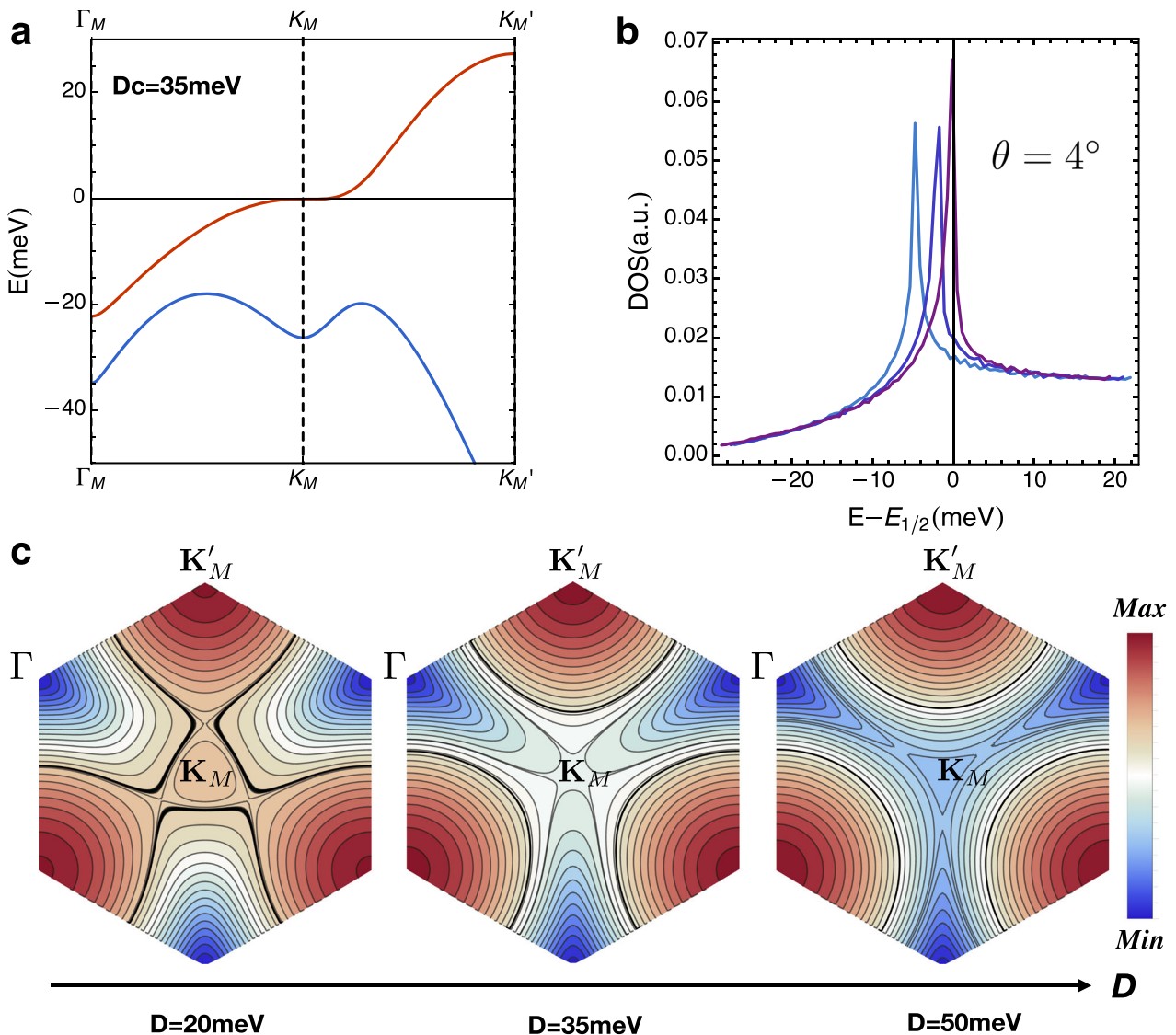

**Fig. 1 Moiré band structures. a** The moiré band structure for one valley of twisted bilayer WSe$_2$. At a critical displacement field $D_c \cong 35$ meV, the $\mathbf{K_M}$ becomes a saddle point of the dispersion. **b** The density of states (with various $D$ near $D_c$) at twist angle $\theta = 4°$. **c** Evolution of energy contours of the topmost moiré band as a function of the displacement field (the dark line corresponds to half-filling). At $D_c$, the system hosts a higher-order van Hove singularity at $\mathbf{K_M}$.

TMDs has a sizable bandwidth comparable to or larger than the interaction energy, which is far from the flat-band limit.

At $\alpha = 0$, the low-energy dispersion has a perfect nesting condition: $\varepsilon_+(\mathbf{k}) = -\varepsilon_-(\mathbf{k})$, so that within the topmost moiré band occupied states of one valley map onto unoccupied states of the other valley under a shift of momentum by a nesting wavevector $\mathbf{Q} = \mathbf{K_M} - \mathbf{K'_M}$. As a result of this perfect nesting, Coulomb interaction—attractive between electrons and holes—immediately leads to an excitonic instability which pairs electron from one valley and holes from the other, namely an intervalley exciton condensate with an order parameter

$$\Delta \sim -\left\langle \sum_{\mathbf{k}} c^\dagger_{\mathbf{k},+} c_{\mathbf{k},-} \right\rangle. \tag{3}$$

The ordered state is fully gapped and electrically insulating. It spontaneously breaks the spin-valley $U(1)_\nu$ symmetry, time-reversal symmetry, and translational symmetry. Therefore, our excitonic insulator is both an electron–hole superfluid at finite commensurate momentum[48] and a spin density wave. Related

intervalley density wave states have also been considered in other moiré systems[49–51].

We can draw an analogy between the intervalley exciton condensate and the BCS superconductivity. After a particle-hole transformation on one of the valleys, the intervalley exciton order parameter becomes precisely an s-wave superconductivity with the valleys playing the roles of spins. We can derive a similar self-consistent equation for the exciton order parameter (see "Methods"). Despite the similarities with the BCS problem, the proximity of hVHS introduces interesting new features for the exciton condensate.

Consider the case with $\alpha = 0$. In the weak coupling regime, we can analytically solve the analogous gap equation as follows,

$$\begin{aligned}\frac{1}{V} &\cong \int_{-\infty}^{\infty} dE \rho(E) \frac{1}{2\sqrt{E^2 + \Delta^2}} \\ &\sim \int_0^{\infty} dE \frac{1}{|E|^{\frac{1}{3}}} \frac{1}{2\sqrt{E^2 + \Delta^2}} \sim 1/\Delta^{1/3}\end{aligned} \tag{4}$$

where we have used the power-law density of the state near the hVHS. We find the intervalley exciton order parameter scales as $\Delta \sim V^3$. In mean-field theory, $T_c^{MF}$ is proportional to the gap, therefore $T_c^{MF} \sim V^3$. This power-law scaling of the critical temperature is distinct from the BCS formula where $T_c \sim \exp(-\frac{1}{VN(0)})$.

Next, we consider the intermediate to strong-coupling regime where the electron and hole form tightly bound pairs, analogous to the BEC limit of Fermi gas. Denoting the UV cutoff in reciprocal space $\Lambda$ and the corresponding bandwidth $2\Lambda^3 = W$, we solve the gap equation for interaction strength $V \sim O(W)$ and find the order parameter scales as $\Delta \sim V$. However, instead of being determined by the pairing gap, the transition temperature is set by the BEC temperature of the exciton gas, which scales as $T_c \sim W^2/V$, proportional to the inverse mass of the excitons[52,53]. An interesting feature is that the crossover scale between weak and strong-coupling behaviors is relatively small in this model due to the presence of the hVHS.

We note that in the strong-coupling flat-band limit, recent works[44,45] have proposed a valley-polarized state that is competitive in energy and distinct from the intervalley exciton condensate proposed here. Experimentally, these two states can be very easily distinguished by their response to an out-of-plane magnetic field. The valley-polarized state features spontaneous magnetization, hysteretic behavior, and (quantized) anomalous hall effect, and it is stabilized by the field. On the contrary, the intervalley exciton insulator is weakened under the field and eventually transitions into the valley-polarized insulator above a critical field.

**Role of various perturbations**. The energy of van Hove singularities relative to the Fermi level at half-filling $\mu$ and the quadratic term in energy dispersion $\alpha$ can be viewed as perturbations to the ideal limit of perfect nesting. Experimentally, the displacement field $D$ tunes both $\alpha$ and $\mu$. Both perturbations have pair-breaking effects on exciton condensation as they lift the degeneracy between electrons in one valley and holes in the other. In particular, under the aforementioned particle-hole transformation, $\mu$ precisely maps to the Zeeman field in a superconductor. We also consider an out-of-plane magnetic field that splits the spin/valley degeneracy by Zeeman energy[54,55],

$$H_B = \pm B_\perp \int_{\mathbf{k}} c_{\mathbf{k},\pm}^\dagger c_{\mathbf{k},\pm}. \tag{5}$$

Due to the orbital angular momentum in TMD systems, the holes in the valence band have large renormalized $g$-factor[55]. Therefore, the primary effect of a weak magnetic field is Zeeman spin/valley splitting. In contrast to $\alpha$ and $\mu$, the Zeeman coupling maps to the chemical potential in a superconductor and its effect is non-pair-breaking. The correspondence between exciton condensate and the superconductor is summarized in Table 1.

In the following, we consider a mean-field theory for the intervalley exciton condensate that includes these perturbations.

The mean-field hamiltonian reads

$$H_{MF} = \sum_{\mathbf{k},\nu=\pm} (\varepsilon_\nu(\mathbf{k}) + \nu B_\perp - \mu) c_\nu^\dagger c_\nu + \Delta c_-^\dagger c_+ + h.c. + \frac{|\Delta|^2}{V}, \tag{6}$$

where $\Delta$ is the order parameter for the exciton condensate and $V$ is the effective interaction strength in the $s$-wave channel. The quasi-particle spectrum is given by

$$E_\pm(\mathbf{k}) = \alpha \mathbf{k}^2 - \mu \pm \sqrt{(\xi(\mathbf{k}) + B_\perp)^2 + |\Delta|^2}. \tag{7}$$

The quasi-particle gap between the two bands is given by $\Delta_g = 2\Delta - \alpha \Lambda^2$, which is an indirect gap for $\alpha \neq 0$.

We calculate the mean-field free energy and vary it with respect to $\Delta$ to arrive the gap equation (see "Methods"),

$$\frac{1}{V} = \sum_{\mathbf{k}} \frac{1}{2\sqrt{(\xi(\mathbf{k}) + B_\perp)^2 + |\Delta|^2}} (n_F(E_-(\mathbf{k})) - n_F(E_+(\mathbf{k}))), \tag{8}$$

where $n_F(E)$ is the Fermi–Dirac distribution. Figure 2c–e shows the quasi-particle gap $\Delta_g$ as a function of $\alpha$, $T$, and $B_\perp$ at coupling $V = 0.1\,W$ (see "Methods" for results with $V = 0.5\,W$). Notice the gap equation is invariant under $\alpha \to -\alpha$, $\mu \to -\mu$. Thus, we only plot for $\alpha > 0$.

First, Fig. 2c shows the quasi-particle gap $\Delta_g$ as a function of $\alpha$ at half-filling at $T = 0$ and $B_\perp = 0$ for $V = 0.1\,W$. With the realistic bandwidth $W \cong 100$ meV, the maximal mean-field quasi-particle gap is $\Delta_g \cong 3.5$ meV, which is close to the activation gap fitted from transport experiments[18]. We note that the intervalley exciton order parameter could survive away from half-filling. We call such state as the excitonic metal (see "Methods" for mean-field phase diagram as a function of $\alpha$ and $\mu$).

Remarkably, the excitonic insulator only exists when the van Hove singularity is tuned close to the Fermi level by the displacement field. For $V = 0.1\,W$, a small detuning in $\mu$ of about $0.01\,W$ (see Fig. 2d) is sufficient to drive the excitonic insulator to a normal metal through a first-order phase transition, which precisely corresponds to the pair-breaking transition of an $s$-wave superconductor driven by the Zeeman field.

Next, we plot the quasi-particle gap as a function of temperature $T$ at half-filling in Fig. 2e. The finite temperature metal–insulator transitions are continuous. For $W = 100$ meV, the maximal critical temperature is $T_c \cong 10$ K, consistent with the onset temperature of insulating behavior[18]. Moreover, the electronic compressibility at half-filling will generically show interesting temperature dependence—as the temperature decreases it first increases due to the divergent DOS and then drop to zero after the critical temperature due to the onset of the insulating gap (see Fig. 2b and "Methods").

Finally, we plot the gap as a function of the out-of-plane magnetic field $B_\perp$ at $T = 0$ and $\mu = 0$ shown in Fig. 2f. We notice the critical field for small $\alpha$ (i.e., good nesting) is much larger than the critical chemical potential. In addition, for all $\alpha$, the quasi-particle gap decreases slowly with $B_\perp$ in the weak field regime, which indicates the effect of the out-of-plane magnetic field is

**Table 1 The correspondence between intervalley exciton condensation and the BCS superconductivity.**

|  | Superconductivity | Intervalley exciton order |
|---|---|---|
| Order parameter | $\Delta_{BCS} \sim c_\uparrow^\dagger c_\downarrow^\dagger$ | $\Delta_{IVE} \sim c_+^\dagger c_-$ |
| Broken symmetries | Charge conservation $U(1)_C$ | Spin-$S_z$ or valley conservation $U(1)_V$ |
| Pair-breaking | Magnetic field $B$ | Chemical potential $\mu$ and displacement field $\alpha$ |
| Non-pair-breaking | Chemical potential $\mu$ | Perpendicular magnetic field $B_\perp$ |

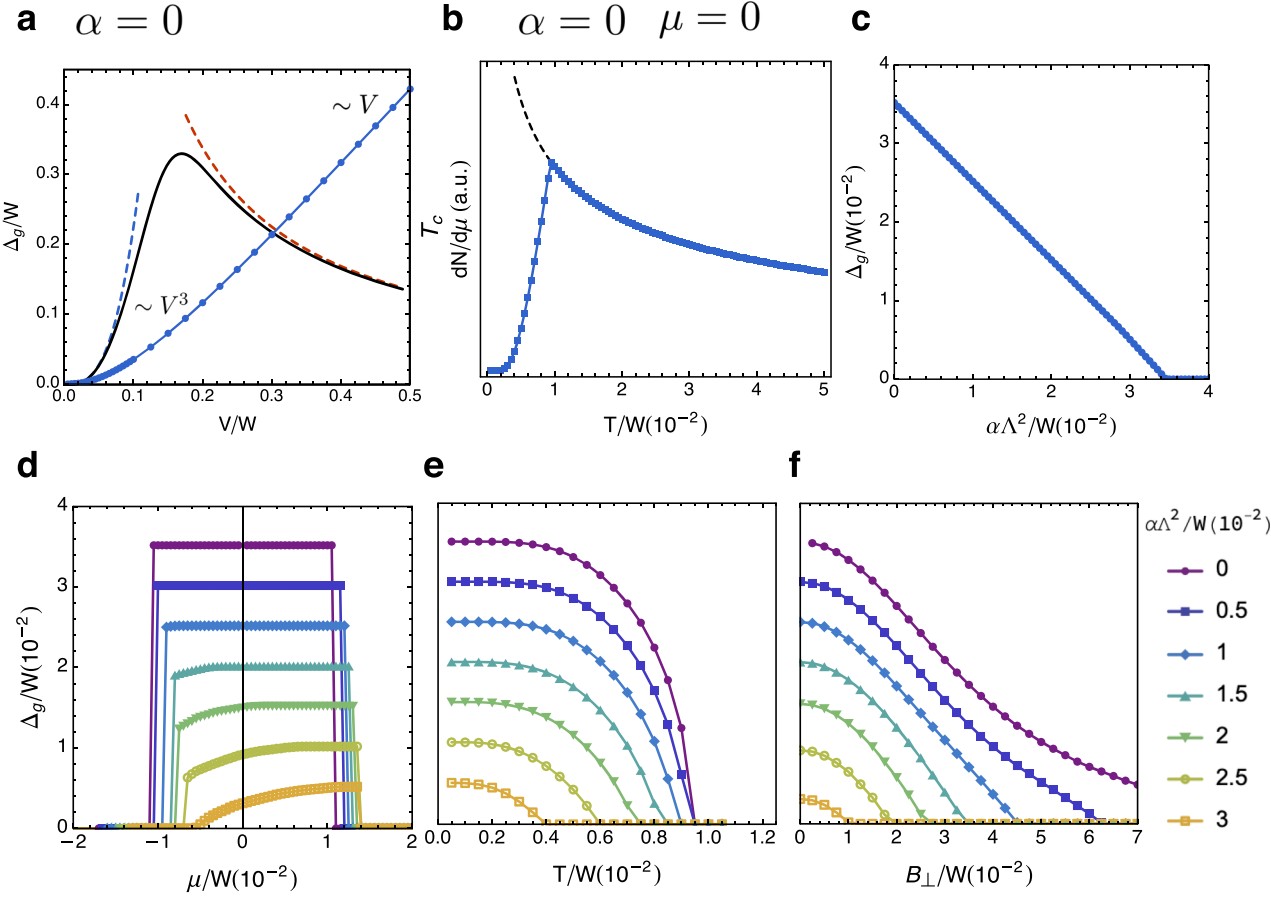

**Fig. 2 The intervalley exciton order. a** The mean-field exciton order parameter (blue) and the schematic plot of $T_c$ (black) as a function of interaction strength for the case of $\alpha = 0$. **b** The mean-field electronic compressibility $dN/d\mu$ as a function of temperature at $\alpha = 0$ and $\mu = 0$. **c** The quasi-particle gap as a function of $\alpha$ with $T = 0$ and $B_\perp = 0$ at half-filling. **d** The quasi-particle gap as a function of $\mu$ with $T = 0$ and $B_\perp = 0$. **e** The quasi-particle gap as a function of $T$ with $B_\perp = 0$ at half-filling. **f** The quasi-particle gap as a function of $B_\perp$ with $T = 0$ and $\mu = 0$. Taking the valley Zeeman g-factor to be ~10[55], the range of magnetic field in (**f**) is 0 ~ 12 T, which is comparable to the experimental available range. All data in (**b**–**f**) are obtained with $V = 0.1\,W$.

non-pair-breaking to the leading order. For larger $B_\perp$, the quasi-particle gap is reduced in an approximately linear way. Furthermore, the orbital effect becomes important in the high-field regime, which is beyond the scope of this work. For a nearly high-order van Hove singularity (i.e., small $\alpha$), the upper critical magnetic field is significantly larger than the critical displacement field, when measured in terms of the Zeeman energy and detuning energy, respectively.

In summary, our theory based on a van Hove singularity near the Fermi level gives an insulating gap and critical temperature comparable to the experimental values and explains the remarkable sensitivity of the gap to the displacement field as well as the lack thereof to the Zeeman field in terms of pair-breaking versus non-pair-breaking perturbation to exciton condensation. In the following, we propose an experiment to directly probe the macroscopic intervalley coherence in the insulating state.

## Discussion

In our theory, the half-filling insulator in TMD homobilayer spontaneously breaks the spin/valley $S_z$ conservation in a similar way that a superconductor spontaneously breaks charge conservation. Therefore, it can be regarded as a spin superfluid that can enable coherent spin transportation over long distances. Moreover, spin polarizations have been shown to be easily

generated and probed via circularly polarized light in TMDs due to the spin-valley locking and valley-selective coupling to chiral photons[17,20–23]. In particular, ref. [23] and ref. [17] demonstrate that in TMD heterostructure there is a nearly perfect conversion from optically generated chiral excitons to spin/valley-polarized holes, as well as a spin diffusion length on the order of 10–20 μm, much longer than the wavelength of the pump/probe light. Therefore, we anticipate that twisted TMDs can provide an ideal platform for studying spin superfluid transport with a fully optical setup.

We propose the following experiments (see Fig. 3a) to detect spin superfluidity in the insulating state of bilayer TMDs. We first create a local spin polarization by circularly polarized light and then use the spatial–temporal resolved circular dichroism spectroscopy[17,23] to monitor its propagation as a function of time (In the experiment, one can add an additional layer of WS$_2$ intentionally misaligned with the twisted bilayer WSe$_2$ in order to enhance the conversion rate of spin/valley polarization from optical pumping[17,23].). In the spin superfluid state, the local spin polarization will propagate ballistically and coherently via collective modes (in the absence of dissipation, see below)[34]. Such ballistic spin transport is a key feature of our intervalley excitonic insulator. In contrast, spins should have diffusive dynamics[34] if the half-filling insulator is valley polarized.

In a spin superfluid, the spin transport equation involves the superfluid phase $\varphi$ that specifies the angle of the in-plane

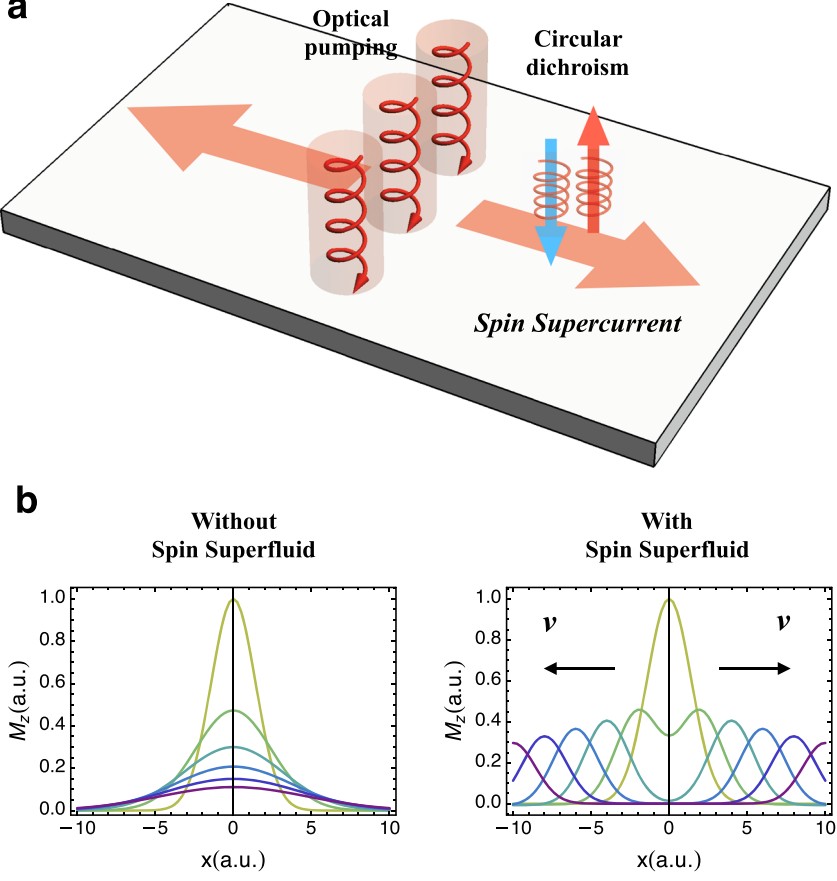

**Fig. 3 Spin transport measurement. a** Setup for all-optical spin transport measurement. (**b**) shows the dynamics of a spin-valley wavepacket in the case with/without spin superfluid (colored curves represent different time instant). (**b**) is obtained by solving the Eqs. (9) and (10) in a 1*d* system with an initial Gaussian distribution of $M_z$ at the origin. The parameters we use in the superfluid case are $v = \sqrt{\rho/K} = 1$, $\tau = 10$. In the diffusive case, we use $D = 1$, $\tau = 10$.

magnetic order parameter[34],

$$\frac{\mathrm{d}M_z}{\mathrm{d}t} = -\nabla \cdot \mathbf{J}_z - \frac{M_z}{\tau}, \tag{9}$$

$$\frac{\mathrm{d}\varphi}{\mathrm{d}t} = -\frac{1}{K}M_z + ..., \tag{10}$$

where $M_z$ is the magnetization along $z$ direction and a conjugate variable to $\varphi$. The spin current $\mathbf{J}_z$ is given by $\mathbf{J}_z = \rho\nabla\varphi$, where $\rho$ is the superfluid stiffness. The parameters $\rho$ and $K$ can be obtained by considering the effective low energy of the Goldstone mode; see the section "Low energy effective theory for the spin superfluid state" under "Methods". $\tau$ is the spin relaxation time, which is expected to be long in TMDs since a spin–flip requires intervalley scattering due to spin-valley locking. In a spin superfluid, the spin-valley wavepacket shows ballistic propagation at the spin-wave velocity $v = \sqrt{\rho/K}$. In contrast, in the absence of spin superfluidity, the spin dynamics is diffusive. The two cases yield completely different transport behaviors as shown in Fig. 3b.

## Methods

**Moiré band structure.** In this section, we show more detailed results on the moiré band structure. We follow the continuum model in ref. [14] to calculate the band structure of twisted bilayer WSe$_2$. The parameters used in this paper are extracted from the first principle calculation. Our continuum model parameters are given as the following: the interlayer tunneling $w = 18$ meV, the intralayer potential $(V, \psi) = (7.9$ meV, $142°)$, and the effective mass given by $m^* = \Delta_g/(2v_F^2)$, where $\Delta_g = 1.6$ eV and $v_F/a = 1.1$ eV.

In Fig. 4a, we show some examples of moiré band structures for different values of the displacement field. The whole dispersion of the moiré band is sensitively dependent on the displacement field.

The band dispersion near $\mathbf{K_M}$ point can be approximated by the form in Eq. (1) in the main text. The coefficient $\alpha$ will be a function of the displacement field. Here, we fit the dispersion near $\mathbf{K_M}$ point to obtain the coefficient $\alpha$ as a function of the displacement field shown in Fig. 4b.

Another important quantity we can extract is the energy distance between the van Hove singularity and the half-filling fermi level. To obtain this quantity, we have to take into account the dispersion in the whole moiré Brillouin zone. We plot this quantity as a function of the displacement field in Fig. 4c. An interesting feature is that, once the van Hove singularity approaches the Fermi level, the Fermi level is more or less fixed near the van Hove energy due to the diverging density of the state.

**Band topology and Berry curvature.** We can calculate the Chern number of the moiré bands with the continuum model. We map the Chern number of the topmost moiré band as a function of twist angle and displacement field. With our model parameters, as shown in Fig. 5, the topmost moiré band has a nontrivial Chern number in the physically relevant regime. Based on these observations, in the strongly coupled regime at half-filling the system likely will enter a valley-polarized phase with quantized anomalous hall response[44,45]. This state is distinct from the intervalley exciton condensate found in the weak coupling limit. For such state, the insulating gap should increase under a perpendicular magnetic field, which is not consistent with the experimental observations.

We can also plot the distribution of the Berry curvature in the moiré Brillouin zone. As shown in Fig. 6, the Berry curvature is mostly concentrated near the $\Gamma$ point in the Brillouin zone for weak displacement field and gradually shifts to $\mathbf{K_M}$ with increasing displacement field. This is expected because the smallest gap between the topmost and second moiré bands is located near $\Gamma$ as shown in Fig. 4a. Near the half-filling fermi surfaces (shown in Fig. 1c), there is essentially zero Berry curvature. Together with the large bandwidth compared to the interaction, we expect the topological property of the band is not essential to the low-energy physics and the weak coupling approach is a good approximation.

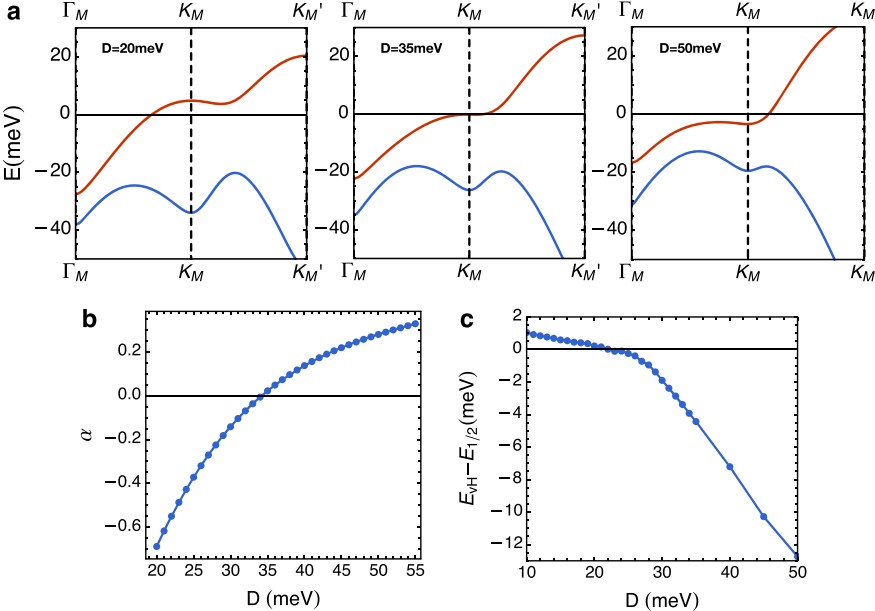

**Fig. 4 Details of the moiré band structure. a** A few examples of the band structure as function at different displacement fields. **b** The phenomenological parameter $\alpha$ as a function of the displacement field obtained by fitting the band structure near $\mathbf{K_M}$ point. The critical displacement field $D_c$ is $\cong 35$ meV. **c** The energy distance between the van Hove singularity and half-filing Fermi level as a function of the displacement field.

## Mean-field gap equation and phase diagram.

Now let us derive the mean-field gap equation. We can obtain the mean-field hamiltonian by Hubbard–Stratonovich transformation of the interacting hamiltonian. Starting from that, we calculate the free energy of the system as the following,

$$F_{MF} = \sum_{\mathbf{k},\nu=\pm} -\frac{1}{\beta}\log\left(1 + e^{-\beta E_\nu(\mathbf{k})}\right) + \frac{|\Delta|^2}{V}, \tag{14}$$

where $E_\pm$ are given in the main text. We can take $\Delta$ to be real and vary the free energy with respect to it,

$$\frac{\partial F_{MF}}{\partial \Delta} = 2\frac{\Delta}{V} + \sum_{\mathbf{k},\nu=\pm} \frac{\Delta}{\sqrt{(\xi_\mathbf{k} + B_\perp)^2 + \Delta^2}} \frac{\nu}{1 + e^{\beta E_\nu}}. \tag{15}$$

Setting the variation to zero, we get precisely the gap equation in Eq. (8) of the main text.

Now we plot the zero-temperature mean-field phase diagram as a function of $\mu$ and $\alpha$ for $V = 0.1\,W$ and $V = 0.5\,W$ in Fig. 7. There are in general three phases in the mean-field phase diagram. For small $\alpha$ and $\mu$, we get the exciton insulator, which has a nonzero exciton condensate and a fully gapped spectrum. For larger $\alpha$, the system develops fermi surfaces while maintaining the exciton order, which we denote as the exciton metal phase. Further increasing $\alpha$ or $\mu$ destroys the exciton order and leaves the system a normal metal. Notice the phase diagram has a strong particle-hole asymmetry induced by finite $\alpha$. We can focus our attention on the half-filling state. In the weak coupling case, as we increase $\alpha$, the system will go from an excitonic insulator to a normal metal through an intermediate phase-separation regime. In the strong-coupling limit, raising $\alpha$ can drive the system from an excitonic insulator to a normal metal through an intermediate exciton metal phase.

In Fig. 8, we plot the correlated insulating gap at half-filling as a function of parameter $\alpha$ for various interaction strength. We also provide the experimental value of the gap as a guide in the plot. We should bear in mind that simple mean-field treatment usually exaggerates the size of the order parameter. At the same time, the transport experiment usually gives a smaller gap due to disorder averaging. In reality, the interaction $V$ is probably closer to 0.3 W. However, one should also take into account the fact that the nesting condition is never perfect which corresponds to finite $\alpha$ (and higher-order terms in the dispersion which become important for large momentum away from the van Hove singularities). For example, if we take $V = 0.3\,W$ and $\alpha = 0.17$, we find the mean-field gap is close to the experimental value. A quantitative comparison between experiments and theory would demand a more sophisticated study of the band structure using, for example, large-scale DFT which is beyond the simple model of this paper. Our model is aimed at understanding the nature of the correlated insulating state based on the experimental observations.

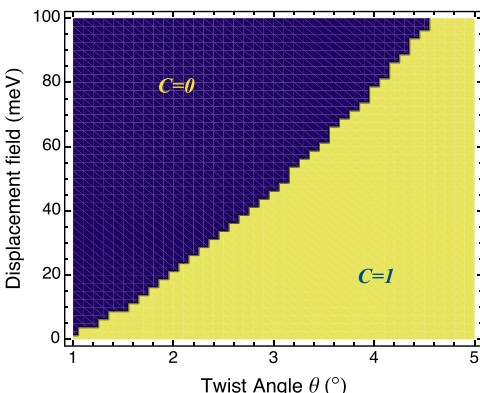

**Fig. 5 Band topology.** The plot shows the Chern number of the topmost moiré band as a function of twist angle and displacement field. The relevant regime in our considerations has nontrivial Chern number.

## Particle-hole transformation and mapping to BCS problem.

Let us make the connection with BCS problem explicit. To that end, we can do a particle-hole transformation to the electron operators in one of the two valleys, for example, the $-$ valley. We define new fermion variables

$$\tilde{c}_{\mathbf{k},+} = c_{\mathbf{k},+}, \quad \tilde{c}_{\mathbf{k},-} = c_{\mathbf{k},-}^\dagger. \tag{11}$$

In terms of these new fermionic operators, the dispersion in Eq. (1) now reads

$$\bar{\varepsilon}_\pm(\mathbf{k}) = \pm(\alpha\mathbf{k}^2 + \mu) + \xi(\mathbf{k}), \quad \xi(\mathbf{k}) = k_y^3 - 3k_y k_x^2, \tag{12}$$

here we also put the chemical potential in the equation. In this new basis, $\xi(\mathbf{k})$ serves as the bare dispersion of the $\tilde{c}$ fermions—it is the same for both the $\pm$ valleys. On the contrary, the $\alpha\mathbf{k}^2 + \mu$ acts like a Zeeman field between the two valleys. In addition, the Coulomb interaction, being repulsive between the original electrons, are now attractive between the $\tilde{c}$ fermions from the two valleys. Therefore, it could promote superconducting states of the $\tilde{c}$ fermions.

After the particle-hole transformation, the intervalley exciton order parameter maps to

$$\left\langle \sum_\mathbf{k} c_{\mathbf{k},+}^\dagger c_{\mathbf{k},-} \right\rangle \rightarrow \left\langle \sum_\mathbf{k} \tilde{c}_{\mathbf{k},+}^\dagger \tilde{c}_{\mathbf{k},-}^\dagger \right\rangle, \tag{13}$$

which is precisely a Cooper pair order parameter. Therefore, the intervalley exciton state is equivalent to a BCS superconductor under the particle-hole transformation to the electrons in one of the two valleys.

## Electron compressibility.

We also study the compressibility of electrons as a function of temperature at half-filling. We calculate the compressibility as the

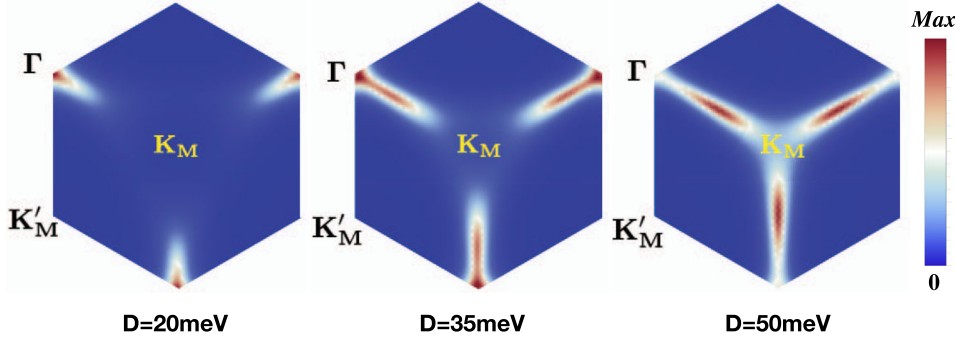

**Fig. 6 Berry curvature.** The plot shows the distribution of the Berry curvature of the topmost moiré band. The calculation is done with a twist angle 4°. Comparing with Fig. 1c, one can see that the half-filling fermi surface is mostly in the regime of essentially zero Berry curvature.

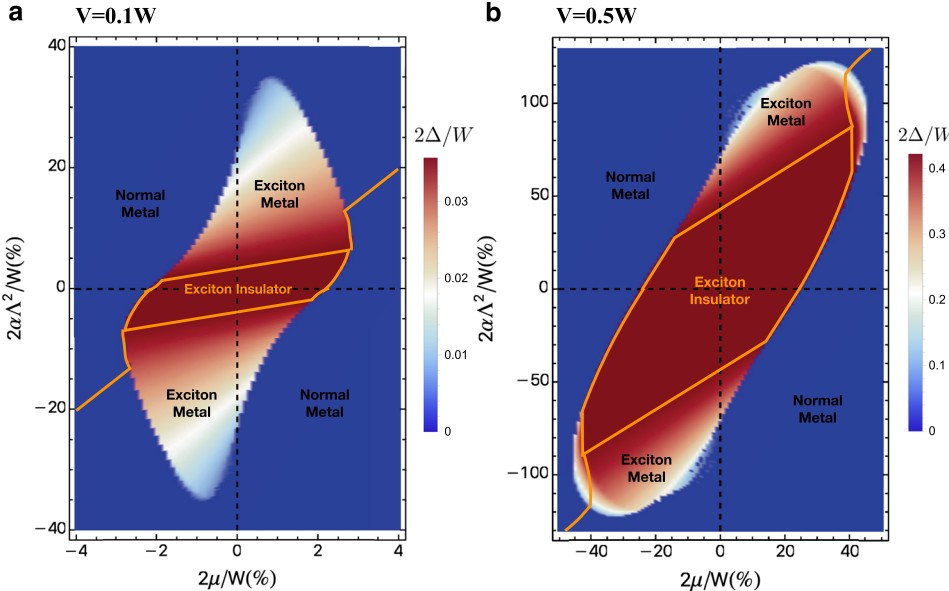

**Fig. 7 Mean-field phase diagram.** The plot shows the zero-temperature mean-field phase diagram - the intervalley exciton order parameter $\Delta$ as function of the $\mu$ and $\alpha$ for **a** $V = 0.1$ W and **b** $V = 0.5$ W. The yellow contour is the phase boundary of the exciton insulator as well as the contour for half-filling.

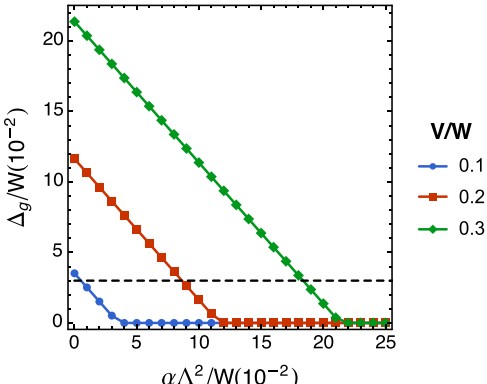

**Fig. 8 Mean-field insulating gap.** The plot shows the insulating gap at half-filling as a function of parameter $\alpha$ for various interaction strength. The dotted line is the observed insulating gap in the experiment (assuming the bandwidth is ~100 meV).

following. The electron number is given by

$$N(\mu, T) = \int dE \rho(E) n_F(E) = \sum_{\nu, \mathbf{k}} \frac{1}{1 + e^{(E_{\nu, \mathbf{k}} - \mu)/T}}. \quad (16)$$

The compressibility is then

$$\frac{dN}{d\mu} = \sum_{\nu, \mathbf{k}} \frac{1}{T} \frac{e^{(E_{\nu, \mathbf{k}} - \mu)/T}}{\left(1 + e^{(E_{\nu, \mathbf{k}} - \mu)/T}\right)^2} \\ - \frac{1}{T} \frac{e^{(E_{\nu, \mathbf{k}} - \mu)/T}}{\left(1 + e^{(E_{\nu, \mathbf{k}} - \mu)/T}\right)^2} \frac{dE_{\nu, \mathbf{k}}}{d\Delta} \frac{d\Delta}{d\mu}. \quad (17)$$

We use the solution of the gap equation $\Delta(\alpha, \mu, T)$ to numerically calculate the compressibility as a function of temperature. A particular case that is easy to handle is at $\alpha = 0$ and $\mu = 0$ because $d\Delta/d\mu$ vanishes due to the particle-hole symmetry. The result for this simple case is shown in Fig. 2b.

Intuitively, there should be two regimes. At low temperature, the intervalley exciton order leads to gapped spectrum—the compressibility should have activation form. The compressibility will increase until the critical temperature of the intervalley exciton order. Then the compressibility is expected to follow certain characteristic power-law decay of the temperature due to the power-law-divergent density of states of the underlying supermetallic state. The result of the numerical calculation of electron compressibility indeed shows these features in Fig. 2b. Most importantly, these features can be readily verified in capacitance measurements.

**Low-energy effective theory for the spin superfluid state.** We derive the effective theory for the phase fluctuation of the exciton order parameter in the weak coupling limit. The effective theory will give us the velocity of the spin-wave excitations. After a Hubbard–Stratonovich transformation, the interacting fermion theory can be brought into the following form (in imaginary time),

$$\mathcal{L} = \sum_{\nu = \pm} \psi_\nu^\dagger (-i\omega + H_{\mathbf{k}, \nu}) \psi_\nu + \Delta \psi_-^\dagger \psi_+ + h.c. + |\Delta|^2/V. \quad (18)$$

Let us assume the exciton order is fixed at $\Delta = |\Delta| e^{i\varphi}$ with $\varphi = 0$. The fermion

green's function is given by

$$\mathcal{G}_\psi = (-i\omega\tau^0 + H_\mathbf{k} + |\Delta|\tau^1)^{-1}. \tag{19}$$

where $\tau$'s are the Pauli matrices in valley space. Consider the effective action for phase fluctuations. The self-energy for $\varphi$ is given by the standard bubble diagram,

$$\Pi(i\Omega, \mathbf{p}) = \int_{\omega, \mathbf{k}} -|\Delta|^2 \, \mathrm{Tr}\left[\tau^2 \mathcal{G}_\psi(i\omega, \mathbf{k})\tau^2 \mathcal{G}_\psi(i(\omega+\Omega), \mathbf{k}+\mathbf{p})\right], \tag{20}$$

For the case of $\alpha = 0$, expanding the self-energy to the second-order in frequency and momentum, we find

$$\Pi(i\Omega, \mathbf{p}) = -c_0|\Delta|^2 - c_1|\Delta|^{-1/3}\Omega^2 - c_2|\Delta|\mathbf{p}^2 + ... \tag{21}$$

where $c_0 = 1/V$ precisely from the mean-field equation, $c_1$ and $c_2$ are constants obtained from convergent integral. In a RPA approximation, the green's function for $\varphi$ is given by

$$\begin{aligned}
G_b(i\Omega, \mathbf{p}) &= \frac{V/|\Delta|^2}{1 + V/|\Delta|^2 \, \Pi(i\Omega, \mathbf{p})} \\
&= -\frac{1}{c_1|\Delta|^{-1/3}\Omega^2 + c_2|\Delta|\mathbf{p}^2}.
\end{aligned} \tag{22}$$

Therefore the effective action for $\varphi$ is

$$\mathcal{L}_{eff} = \int_{\mathbf{x}, \tau} K(\partial_\tau \varphi)^2 + \rho(\partial_x \varphi)^2, \tag{23}$$

where $K = c_1|\Delta|^{-1/3}$ and $\rho = c_2|\Delta|$. The superfluid velocity is given by $v = \sqrt{\rho/K} = \sqrt{c_2/c_1}|\Delta|^{2/3}$. The case of $\alpha \neq 0$ is more involved and we leave it to future work. However, the form of effective action in Eq. (23) is generally applicable in any superfluid state.

## Data availability
The datasets generated during this study are available on reasonable request.

## Code availability
The custom codes generated during this study are available on reasonable request.

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

## Acknowledgements
We thank Augusto Ghiotto, En-Min Shih, Abhay Pasupathy, and Cory Dean for sharing their experimental results prior to publication, and Yang Zhang for sharing the first principle data and collaboration on related work. We thank Ran Cheng, Feng Wang, Noah F. Q. Yuan, Yi-Zhuang You, and Shu Zhang for useful discussion. This work is supported by DOE Office of Basic Energy Sciences under Award DE-SC0018945. Z.B. is supported by the Pappalardo fellowship at MIT, partially by KITP program on the topological quantum matter under Grant No. NSF PHY-1748958, and by the startup funding from the Pennsylvania State University. L.F. is partly supported by Simons Investigator Award from the Simons Foundation.

## Author contributions
Z.B. and L.F. both contributed essentially to the original idea, model calculation, theoretical analysis, and paper preparation.

## Competing interests
The authors declare no competing interests.
