## [Peer Review File · Nature Communications]

Editorial Note: Parts of this peer review file have been redacted as indicated to remove third-party material where no permission to publish could be obtained.

REVIEWER COMMENTS

Reviewer #1 (Remarks to the Author):

This is a very timely work which studies the nature of the insulating state observed in twisted bilayer transition metal dichalcogenides. The authors suggested that the experimentally observed insulating phase near 4 degree of twisted angle, which is very sensitive to displacement field, is indeed an excitonic density wave formed by the pairing of electrons and holes in mini-bands at different valleys. They pointed out that the higher order von Hove singularity of the bands plays an important role in enhancing the excitonic condensation. These results are interesting and timely.

However, I have a few potentially serious concerns and reservations as listed below:

1. The authors claimed that they can explain the magnetic field dependence of the suppression of the insulating gap. However, the authors only considered the Zeeman effect of the perpendicular magnetic field. When the magnetic field is as large as several Teslas, the orbital effects would dominate. Experimentally, Landau levels were observed in Ref.1. Therefore, it is hard to relate the results of the authors to the experiment as the authors only considered the Zeeman effect.

2. Concerning the optical detection of the spin superfluid state, the authors proposed to generate local spin polarization through a circularly polarized light and then use spatial-temporal resolved circular dichroism spectroscopy for the detection of the diffusion of the spin polarization. In monolayer transition metal dichalcogenides, light excite electrons from the valence bands to the conduction bands and the relevant energy scale is in the order of eV. In the current proposal, the relevant energy scale (such as the excitonic insulator gap) is in the order of a few meV. How can ordinary optical method be used without generating a large number of unwanted excitations?

3. For the detection using circular dichroism spectroscopy, the spatial resolution of the detection is limited by the wavelength of light which is at least in the order of several hundreds of nanometers even using visible lights. How can such a low spatial resolution method be used to detect the proposed spin transport?

The current work is timely and interesting. However, I believe that the authors have not thought through the details of the problem carefully. In its current form, the work falls short of the standard of Nature Communications.

Reviewer #2 (Remarks to the Author):

The authors propose an intervalley excitonic condensate as a candidate for the insulating phase recently observed in twisted bilayer WSe₂. The authors employ a mean field analysis to study such phase arguing, in particular, that the presence of a van Hove singularity close to half-filling drives the instability leading to excitonic pairing. They also discuss the possible response of this phase to different perturbations such as chemical potential or out-of-plane field. In addition, they propose an experimental setting to test their proposed state. The manuscript is well written and addresses an interesting question in the rapidly expanding field of Moire materials. Intervalley excitonic condensates of the type proposed have been studied in the context of graphene Moire materials starting from the early works, Refs. 46, 47, 48, and more recently in arxiv:1901.08110 and

1911.02045. The main new ingredient here is that, due to spin-orbit coupling, the excitonic intervalley coherent pairing is also associated with a non-trivial spin structure which the authors propose can be detected experimentally.

Below I provide a more detailed point by point criticism:

1. In the introduction, the authors argue that the smallness of the insulating gap rules out a Mott scenario, yet their weak coupling estimate for the gap scales as the third power of the interaction which will generally be quite large. They argue that in intermediate to strong coupling, the gap scale is instead given by W^2/V but then they mention that in this case the phase is actually a Mott phase. In addition, all their calculations are performed in the weak coupling limit $V=0.1W$ which is relatively far from the relevant experimental scales. The authors should clearly explain whether there is a distinction between their proposed phase and a Mott phase at strong coupling and also comment on the compatibility of the gap scale obtained in their calculation with experiments.
2. The authors do not discuss at all the effects of band topology which may strongly affect the scenario they propose here. In particular, the two valleys are related by time-reversal symmetry so in principle each valley can have a non-vanishing Chern number. This possibility is realized in graphene based Moire materials e.g. twisted bilayer graphene aligned on top of hBN, 1901.08209, 1901.08110 and twisted double bilayer graphene 1903.08685). It has also been discussed in TMD homobilayers (Ref. 15). Even if the total Chern number in each band vanishes the local Berry curvature maybe large. This may be unimportant at weak coupling, but at intermediate coupling relevant to the realistic system, it may have important ramifications for the scenario proposed here since the excitonic pairing would take place between opposite Chern bands making it very different from bilayer quantum Hall systems. In particular, it means that the "superconducting" analog obtained by performing a particle-hole transformation in one valley now corresponds to superconducting pairing in a band with non-vanishing Chern number which is generally not favored. The effect of band topology on excitonic intervalley coherent pairing was discussed in arxiv:1901.08110 in the context of twisted bilayer graphene on top of hBN substrate. The authors should include a detailed discussion of the global and local band topology and explain how intervalley coherent pairing can survive in the current context.
3. The analysis of the authors relies on the existence of van Hove singularities close to half-filling which was derived based on the non-interacting band structure. However, this band structure is likely to receive interaction renormalization effects coming from the remote bands which is known to be an important effect in graphene-based Moire systems (see 1907.11723, 1812.04213, 1911.02045). The authors should justify why using the van Hove singularities arising from the non-interacting band structure is expected to yield quantitatively correct result. Is there any symmetry reason to expect the form of the dispersion in Eq. 1 to survive in the presence of interaction renormalization effects? Or are the authors arguing that there exists some displacement field for which a van Hove singularity emerges regardless of the band structure details?
4. The authors argue that out-of-plane magnetic field does not act as pair breaking due to Zeeman effect. However, out-of-plane field is also generally expected to couple to orbital degree of freedom leading to a "valley" Zeeman effect which can have a much larger g-factor (See 1908.05110). The authors should comment on whether this effect will lead to pair breaking.

Reviewer #3 (Remarks to the Author):

The authors analyze the properties of AA stacked WSe₂ bilayer moiré crystals with a half-filled hole band. The work was motivated by a recent experiment that reported strong insulating behavior over a surprisingly narrow range of external displacement field vanishes. They interpret this finding as being due to the formation of an excitonic density wave state, arguing that this scenario is more likely since the bands are relatively broad, and predict that it will be possible to

perform an all-optical experiment that demonstrates spin-superfluidity.- which is an expected property of these states. The density wave is stabilized by a nesting condition that is satisfied over a narrow range of displacement fields.

I recommend publication of this manuscript which makes an interesting and testable prediction. The arguments advanced in favor of this prediction are plausible and carefully discussed. I have some suggestions that the author might consider.

- i) The second last paper in the abstract is awkward and needs to be rewritten.
- ii) The authors refer to the state they propose as an excitonic density-wave. It may be that this terminology is perfectly standard - but it seems to me that it requires a few comments. I think that they are saying that their state is a spin and charge density wave state of the type that can be viewed as an exciton condensate. Is there at least a reference that could be cited where the meaning of this state name is carefully defined? In any event, because it is a density wave, collective transport is pinned. The authors do account for relaxation between valleys when they consider transport, but the flip side of this is coupling between the valleys in the ground state. I believe that there are strong arguments related to momentum conservation that these effects are weak - but perhaps this should be discussed explicitly.

Response to Reviewers' comments (NCOMMS-20-23082)

We sincerely thank all three reviewers for taking their time to carefully review our work and prepare very insightful reports. To summarize, all reviewers found the results in our manuscript interesting and timely but asked for more efforts to clarify detailed applicability of our theoretical approach and the feasibility of our proposed optical experiments. We have carefully addressed all the questions/comments in this reply. We have also made changes in the presentation of the main text to make the story as clear as possible. We believe the reviewers' insightful comments have greatly helped to improve the readability and technical robustness of our manuscript. In the following, we prepared a point by point reply. In the following, the red text are the changes in the main text. The corresponding changes are also marked red in the revised manuscript as required.

Referees' comments:

Reviewer #1 (Remarks to the Author):

This is a very timely work which studies the nature of the insulating state observed in twisted bilayer transition metal dichalcogenides. The authors suggested that the experimentally observed insulating phase near 4 degree of twisted angle, which is very sensitive to displacement field, is indeed an excitonic density wave formed by the pairing of electrons and holes in mini-bands at different valleys. They pointed out that the higher order von Hove singularity of the bands plays an important role in enhancing the excitonic condensation. These results are interesting and timely.

However, I have a few potentially serious concerns and reservations as listed below:

We are glad that the reviewer finds our results interesting and timely. We also agree that the issues that are raised by the reviewer below are important and should be better addressed. We provide detailed explanations to each specific question. Guided by these questions/comments, we also improve our presentations in the main text.

1. The authors claimed that they can explain the magnetic field dependence of the suppression of the insulating gap. However, the authors only considered the Zeeman effect of the perpendicular magnetic field. When the magnetic field is as large as several Teslas, the orbital effects would dominate. Experimentally, Landau levels were observed in Ref.1. Therefore, it is hard to relate the results of the authors to the experiment as the authors only considered the Zeeman effect.

As nicely pointed out by the reviewer, the magnetic field would lead to two major effects, namely the Zeeman splitting of the spins and orbital effect which leads to Landau level physics. We completely agree with the reviewer that the orbital effect will have important consequences in high magnetic field. In our manuscript we have only considered the Zeeman splitting for the following reasons.

First of all, due to the angular momentum carried by the valley degree of freedom, the effective g-factor in the TMD system is large. As demonstrated in Ref 49, the effective g-factor can be as

large as 10. Therefore, we expect a much larger energy splitting of the states from two valleys than the ordinary spin Zeeman splitting. Secondly, in the experiment of Ref 1, the correlated insulating state survive up to 5T perpendicular magnetic field. In this range of magnetic field, we can estimate the amount of flux going through each moiré unit cell. For twist angle 4 degree, the moiré unit cell has lattice constant of $\sim 5\text{nm}$ and consequently the magnetic flux through the unit cell is $0.052\phi_0$, which is a small fraction of the magnetic flux quanta. We can do a more quantitative comparison of energy scales. At 5T, the Zeeman splitting is as large as $E_Z = 2g^*\mu_B B \approx 2 * 10 * 5.79 * 10^{-2} * 5 = 5.79 \text{ meV}$ due to the enhanced g-factor. We can also estimate the Landau level splitting due to the orbital effect of the magnetic field, $E_L = \frac{\hbar e B}{m} \simeq 1.3 \text{ meV}$, which is rather small due to the large effective mass of WSe2 hole bands ($m^* \sim 0.44 m_e$). Therefore, the range of magnetic field in experiment can still be viewed as in the weak field regime and we expect the orbital effect to be small compared to that of the Zeeman splitting.

We understand that the calculation of the insulating gap in our treatment is not reliable in the high field regime. However, the statement about the non-pair-breaking effect of the magnetic field is made in weak field regime where the Zeeman field approximation is trustworthy. Therefore, we have added a few sentences stressing this caution and weaken our statement in the high field regime. We add on page 3 section C after Eq. (5):

“Due to the orbital angular momentum in TMD systems, the holes in the valence band have large renormalized g -factor^{TMDgfactor}. Therefore, The primary effect of a weak magnetic field is Zeeman spin/valley splitting. In contrast to α and μ , the Zeeman coupling maps to the chemical potential in a superconductor and its effect is non-pair-breaking.”

On page 4 we add:

“In addition, for all α , the quasiparticle gap decreases slowly with B_{\perp} in the weak field regime, which indicates the effect of out-of-plane magnetic field is non-pair-breaking to the leading order. For larger B_{\perp} , the quasiparticle gap is reduced in an approximately linear way. Furthermore, orbital effect becomes important in high field regime, which is the beyond the scope of this work.”

We also change the sentence about magnetic field effect in the abstract to the following:

“Our theory explains the remarkable sensitivity of the insulating gap to the vertical electric field. In contrast, our theory shows that the insulating gap is reduced mildly by a perpendicular magnetic field, with quadratic dependence at low field. These physics can be understood in terms of pair-breaking versus non-pair-breaking effects in a BCS analog of the system.”

2. Concerning the optical detection of the spin superfluid state, the authors proposed to generate local spin polarization through a circularly polarized light and then use spatial-temporal resolved circular dichroism spectroscopy for the detection of the diffusion of the spin polarization. In monolayer transition metal dichalcogenides, light excite electrons from the valence bands to the conduction bands and the relevant energy scale is in the order of eV. In the current proposal, the relevant energy scale (such as the excitonic insulator gap) is in the order of a few meV. How can ordinary optical method be used without generating a large number of unwanted excitations?

We thank the referee for raising this excellent question. This helps us to reflect on more details of our proposal and make some improvements which we now include in our main text. We translate the reviewer's question into the following. In order for the proposed experiments to work, we have to achieve the following two goals: 1) the efficiency of conversion from optical excitations to the pure valley imbalance should be as high as possible; 2) the density of generated valley imbalanced hole excitation should be small compared to the full-filling density of the moiré sub-band (in order to not overwhelm the moiré physics). Luckily both of the goals have already been achieved in TMD and TMD based moiré system by our experimental colleagues from Berkeley and Cornell (Ref 23 and 18). Therefore, we do think it is conceivable that our proposal (in the modified version) could be successful. In the following, we briefly discuss how the two issues could be resolved in our system.

The first issue about the efficiency of conversion is very critical for the proposed experiment. If one start with a monolayer of WSe₂, unfortunately the conversion rate of optical excitations to pure valley imbalanced hole is low (~ 0.1% to 1% depending on the sample quality see Ref.). Experimentally, this is due to the valley exchange interactions of the excitons which can quickly wash out the valley information of the optically generated excitons before the electrons recombine with the holes. Such an effect indeed is a possible drawback for the proposed experiment in twisted bilayer WSe₂ of our paper. However, this problem has a very nice remedy by using WS₂/WSe₂ heterostructure (Ref 23 and 18). Here we borrow a figure from Ref 23 to explain that the mechanism of enhancing the conversion rate with an addition of WS₂ layer. (A) By pump circularly polarized light, excitons in the K valley of WSe₂ are selectively excited. Before the intervalley exchange happens, an ultrafast interlayer charge transfer process (between WS₂ and WSe₂) takes place and efficiently converts the excitons into excess holes within ~100 fs. (B) Electrons in WS₂ recombine with K valley and K' valley holes in WSe₂ with almost equal probabilities, resulting in an excess of K valley holes and a deficiency of K' valley holes in WSe₂. This process has been demonstrated in experiments to have almost perfect (~100%) conversion of the optical excitations to valley imbalanced holes (Ref. 23) – which is the key condition for the spin dynamics measurement later. Therefore, in our twisted bilayer WSe₂ case, we expect that adding another layer of WS₂ (presumably with a large twist angle to avoid an interfering moiré superlattice) can also greatly enhance the conversion efficiency of valley-imbalanced holes.

[Redacted]

The second issue is easy to fix. With a fixed conversion rate, the density of valley imbalanced hole is essentially determined by the power of light used in the optical pumping. In the existing experiments on WS₂-WSe₂ heterostructure (Ref 23 without moiré superlattice, Ref 18 with moiré superlattice), the optically generated imbalanced hole density can range from $10^{10}/\text{cm}^2$ to $10^{12}/\text{cm}^2$. This actually is at a sweet spot for exploring moiré physics. The full-filling of moiré sub-band in our twisted bilayer WSe₂ case is $\sim 10^{13}/\text{cm}^2$ for twist angle ~ 4 degree. The generated valley imbalanced hole can be indeed viewed as a small perturbation to the moiré physics and will not swamp it.

Perhaps the “unwanted excitations” mentioned in the reviewer’s comment is worrying about that the optically generated valley-imbalanced holes may initially have a broad distribution in energy in the moiré sub-band. We think this will not be the case at least in our setting. Since all the experiments are conducted in at temperature lower than the insulating gap (which is much smaller than the moiré bandwidth $\sim 100\text{meV}$), the distribution of holes will quickly equilibrate within a valley even if the initial distribution is relatively broad.

In summary, we think the current twisted bilayer WSe₂ system (with an additional WS₂ layer) has all the key ingredients to facilitate the measurement of spin dynamics using optical pump-probe scheme.

We have added a few sentences on the mechanism of pure valley polarized hole and the revised proposal (in section III):

“Moreover, spin polarizations have been shown to be easily generated and probed via circularly polarized light in TMDs due to the spin-valley locking and valley-selective coupling to chiral photons. In particular, Ref[23] and Ref[18] demonstrate that in TMD heterostructure there is a nearly perfect conversion from optically generated chiral excitons to spin/valley polarized holes, as well as a spin diffusion length on the order of $10\sim 20\ \mu\text{m}$, much longer than the wavelength of the pump/probe light. Therefore, we anticipate that twisted TMDs can provide an ideal platform for studying spin superfluid transport with a fully optical setup.”

We also include a footnote on the experimental proposal,

“[57] In the experiment, one can add an additional layer of WS₂ intentionally misaligned with the twisted bilayer WSe₂ in order to enhance the conversion rate of spin/valley polarization from optical pumping\cite{optical2019,SpinSF3}.”

We hope these revisions clear things up.

3. For the detection using circular dichroism spectroscopy, the spatial resolution of the detection is limited by the wavelength of light which is at least in the order of several hundreds of nanometers even using visible lights. How can such a low spatial resolution method be used to detect the proposed spin transport?

We thank the reviewer's remark on the length scale of the spin dynamics. However, we think this is not an issue in the current situation. We would like to again refer the reviewer to the existing experiments measuring spin dynamics in TMD (Ref 23) and TMD moiré systems (Ref 18).

In Ref 23 (WSe₂-WS₂ heterostructure *without moiré physics*), measurements of spin diffusion current signal up to the scale of 8 μm are successfully performed – such length scale are in fact one order of magnitude larger than the wavelength of the light applied to generate spin/valley imbalance (the pump-probe light has energy $\sim 1.8\text{eV}$ which has wavelength $\sim 700\text{nm}$). The fact that one can observe such long scale spin diffusion is mainly attributed to the well-preserved valley conservation symmetry – which is also present in our system.

Ref 18 (WSe₂-WS₂ heterostructure *with a long-range moiré superlattice*) finds remarkably that in the half-filling correlated insulating state of a moiré sub-band the spin lifetime is actually enhanced (2~4 times longer) compared to the metallic states. This would warrant a diffusion length on the order of 20 μm – much larger than the wavelength of the light. While the mechanism of enhanced spin lifetime is not fully determined, this gives us further confidence that meaningful measurement could be conducted in the correlated insulating state of our system.

Besides, the above-mentioned systems only have diffusive spin dynamics. Our system, in which we predict to host spin-superfluid state, could potentially support even longer coherent spin transport. Therefore, we think the spatial resolution is not an issue for detecting the spin transport.

The current work is timely and interesting. However, I believe that the authors have not thought through the details of the problem carefully. In its current form, the work falls short of the standard of Nature Communications.

We again thank the reviewer for recognizing our timely and interesting work. The comments/questions helped us greatly improve our manuscript. In the above reply, we address the detailed problems raised by the reviewer point by point and revise the paper accordingly. We hope to convince the reviewer that we have thought through the proposed experiment and that it is indeed feasible in our setting. Hopefully, the revised manuscript can now meet the high standard of Nature Communications.

Reviewer #2 (Remarks to the Author):

The authors propose an intervalley excitonic condensate as a candidate for the insulating phase recently observed in twisted bilayer WSe₂. The authors employ a mean field analysis to study such phase arguing, in particular, that the presence of a van Hove singularity close to half-filling drives the instability leading to excitonic pairing. They also discuss the possible response of this phase to different perturbations such as chemical potential or out-of-plane field. In addition, they propose an experimental setting to test their proposed state. The manuscript is well written and addresses an interesting question in the rapidly expanding field of Moiré materials. Intervalley excitonic condensates of the type proposed have been studied in the context of graphene Moiré materials starting from the early works, Refs. 46, 47, 48, and more recently in arxiv:1901.08110 and 1911.02045. The main new ingredient here is that, due to spin-orbit coupling, the excitonic

intervalley coherent pairing is also associated with a non-trivial spin structure which the authors propose can be detected experimentally.

We thank the reviewer for the very nice summary of our work. In the revised version, we have added some new references including 1901.08110 and 1911.02045. Indeed, moiré materials is a rapidly developing field. We feel obligated to credit relevant works and hope this will better serve the community and advance the field of moiré physics. In the following, we will address the questions/comments from the reviewer point by point.

Below I provide a more detailed point by point criticism:

1. In the introduction, the authors argue that the smallness of the insulating gap rules out a Mott scenario, yet their weak coupling estimate for the gap scales as the third power of the interaction which will generally be quite large. They argue that in intermediate to strong coupling, the gap scale is instead given by W^2/V but then they mention that in this case the phase is actually a Mott phase.

We want to be clear that our claim is in the strongly coupled regime the insulating gap scales linearly with the interaction strength V . However, the ordering temperature of the inter-valley exciton condensate, not the insulating gap, scales as W^2/V . This limit is an analog of the BEC regime in the BCS-BEC crossover. It is also similar to the case of AFM ordering temperature in parent compound of Cuprates. The mott gap is on the order of Hubbard interaction U , however AFM ordering temperature is on the order of the exchange interaction $J \sim t^2/U$. That being said, we must clarify that the curve of T_c in Fig. 2 (a) is not obtained from actual calculation but from the educated guess based on BCS-BEC crossover physics.

In addition, all their calculations are performed in the weak coupling limit $V=0.1W$ which is relatively far from the relevant experimental scales. The authors should clearly explain whether there is a distinction between their proposed phase and a Mott phase at strong coupling and also comment on the compatibility of the gap scale obtained in their calculation with experiments.

We thank the reviewer for raising this point. In our current manuscript, we consider approaching the problem of the correlated insulator from the weak coupling limit, which is motivated by the following considerations. First, the energy scale of the interaction V compared to the bandwidth W is indeed small ($V/W \sim 0.3 - 0.5$) for the large twist angles in the experiment – this is quite different from the case of twisted bilayer graphene system where the interaction V is on the same order of or even larger than W . (Even for TBG, Hartree-Fock mean field study is a common practice within the community, see for example in npj Quantum Materials 4.16, Phys. Rev. Lett. 124, 097601, Phys. Rev. Lett. 124, 166601, Phys. Rev. X 10, 031034, 1911.03760.) In addition, the Mott gap observed in transport experiments is rather small $\sim 3\text{meV}$. Therefore, the system is not deep in the Mott regime, where the Mott gap would be of the same order as V . Next, changing the displacement field or doping the correlated insulator results very quickly in a metallic state, where potential signature of superconductivity is also observed. A weak coupling theory could naturally capture both the correlated insulator and metal phase in the same framework.

In our main text we present mostly the result of $V=0.1W$ case where the mean field approximation is reliable. (Similar mean field calculations with $V=0.5W$ has been presented in the supplementary section.) For $V=0.1W$ in the case of perfect nesting our calculation gives a close number of the insulating gap compared to the experimental results. One should also bear in mind that simple mean field treatment usually exaggerates the size of the order parameter. At the same time, transport experiment usually gives a smaller gap due to disorder averaging. In the real experiments, the interaction V is probably bigger than $0.1W$, closer to $0.3W$. However, one should also take into account the fact that the nesting condition is never perfect which corresponds to finite α (and higher order terms in the dispersion which become important for large momentum away from the van Hove singularities). For example, with $V=0.3W$ and $\alpha=0.17$, we find the mean field gap is $\sim 3\text{meV}$, close to the experimental value. (We add a new paragraph and a plot about the insulating gap as function of α for various interactions in section IV D.) An honest comparison between experiments and theory would demand a more sophisticated study of the band structure using for example large scale DFT which is beyond the scope of the paper. Our simple model is set to provide an intuitive picture for the many-body physics in the system and indeed gives good qualitative results on the insulating gap and other behaviors compared to the experiment.

For the last comment about the distinction between the weak coupling insulating state and the strong coupling state, it is deeply connected to the next question about topology of the relevant moiré band. Therefore, we feel it is more appropriate to include the answer together with the next question.

2. The authors do not discuss at all the effects of band topology which may strongly affect the scenario they propose here. In particular, the two valleys are related by time-reversal symmetry so in principle each valley can have a non-vanishing Chern number. This possibility is realized in graphene based Moire materials e.g. twisted bilayer graphene aligned on top of hBN, 1901.08209, 1901.08110 and twisted double bilayer graphene 1903.08685). It has also been discussed in TMD homobilayers (Ref. 15). Even if the total Chern number in each band vanishes the local Berry curvature maybe large. This may be unimportant at weak coupling, but at intermediate coupling relevant to the realistic system, it may have important ramifications for the scenario proposed here since the excitonic pairing would take place between opposite Chern bands making it very different from bilayer quantum Hall systems. In particular, it means that the “superconducting” analog obtained by performing a particle-hole transformation in one valley now corresponds to superconducting pairing in a band with non-vanishing Chern number which is generally not favored. The effect of band topology on excitonic intervalley coherent pairing was discussed in arxiv:1901.08110 in the context of twisted bilayer graphene on top of hBN substrate. The authors should include a detailed discussion of the global and local band topology and explain how intervalley coherent pairing can survive in the current context.

We thank the reviewer’s insightful comment.

First, with our model parameter, the continuum model calculation shows a nontrivial Chern number for the topmost moiré band. This is consistent with results from Ref. 15. We add the discussion of global topology and the berry curvature distribution in the revised version in the method section as required by the reviewer. In this scenario, the weakly coupling ground state,

the intervalley exciton state we proposed, will be different from the ground state in the strongly coupled regime, which is most likely a valley polarized state. The evolution between the two states will be an interesting problem to study in the future.

We of course are aware that the band topology plays an important role when the system goes into the flat-band limit. As the reviewer nicely point out, in the flat-band limit, the intervalley exciton order might be suppressed due to the winding of order parameter in the momentum space as a result of the non-trivial Chern number. We totally agree that in the flat-band or strong coupling limit, the most proper ground state of half-filling is valley polarized state. This state will have distinct properties that is somewhat easy to spot in experiments. For example, as shown in the ABC tri-layer graphene and TBG systems, valley polarized states with non-trivial chern number will have spontaneous anomalous hall effect with quantized Hall conductance as well as magnetic hysteresis, none of which are observed in the transport experiment in current system. In addition, while the valley polarized state will be stabilized by the out-of-plane magnetic field, the experiment shows that the correlated insulator is destroyed by increasing magnetic field. Therefore, the experiment clearly speaks for a state that is different from which one will get in the strongly coupled regime - this is another motivation for us to adopt the weak coupling approach. In the weak coupling limit, as the reviewer kindly noted, the most relevant factor in determining the ground state order is the information about fermi surface and van Hove singularities, which is the focus of our paper.

We add a **new subsection, section IV B “Band topology and Berry curvature”**, which detailly discusses the global and local topology of our system. In particular, we plot the Chern number of the topmost moiré band as function of twist angle and displacement field based on the continuum model and the distribution of the berry curvature in the moiré brillouin zone (as function of displacement field). We observe that most of the berry curvature are concentrated near the Gamma point of the brillouin zone. This is expected because the smallest gap between the topmost and second moiré bands is located near Gamma. Near the half-filling fermi surface and the van Hove singularities there is essentially zero berry curvature in our continuum model. Therefore, the weak coupling result will not be sensitive to the topological property of the band.

We add the following sentences on page 2 paragraph 2:

“In the relevant parameter regime, the moiré band has non-zero Chern number. Details of the topological properties of the moiré band are discussed in section IV B. While playing an important role in the flatband limit [44-46], the topological property does not appear to play a significant role in the weak coupling scenario considered here.”

We also add the following sentences on page 3 to clarify the distinction of the weak and strong coupling states:

“We note that in the strong-coupling flat band limit, recent works [45,46] have proposed a valley-polarized state that is competitive in energy and distinct from the intervalley exciton condensate proposed here.”

Finally, we will be remiss if we do not mention some later references on the same platform, for example Phys. Rev. Research 2, 033087, which apply a different set of parameters in the continuum model and suggest that the relevant moiré bands have trivial Chern number. In such a scenario with topological trivial moiré band, the ground state in weak coupling will be smoothly connected with the strongly coupled mott insulator without phase transition. We do think our parameters are more appropriate because the band structure obtained in our model is much closer to the band structure from DFT calculation in Ref. 1. However, the physics of (higher order) van Hove singularities are universally observed regardless of which set of parameters is used. Therefore, the prediction of the half-filling intervalley excitonic state is robust.

3. The analysis of the authors relies on the existence of van Hove singularities close to half-filling which was derived based on the non-interacting band structure. However, this band structure is likely to receive interaction renormalization effects coming from the remote bands which is known to be an important effect in graphene-based Moire systems (see 1907.11723, 1812.04213, 1911.02045). The authors should justify why using the van Hove singularities arising from the non-interacting band structure is expected to yield quantitatively correct result.

We are grateful for the reviewer to raise such an excellent point. Indeed, as the reviewer pointed out, for TBG the remote band has important effects in renormalizing dispersions for the moiré bands near charge neutrality. First of all, the remote bands (from both conduction and valence bands) are close in energy with the relevant moiré bands. The energy separations between remote bands and bands near charge neutrality are typically smaller than or on the same order of the interaction scale in TBG. Second, the interaction strength in TBG is mostly larger than the bandwidth. Therefore, when partially filling the moiré bands away from charge neutrality, the flat-band dispersion is wildly modified due to the Hartree correction from charge modulation. For these reasons, it is a bad approximation in TBG to simply consider the middle bands and ignore the remote bands.

We are aware of the renormalization effects of the remote bands in TBG. However, we expect the renormalization in the tWSe₂ case to be much smaller than the graphene-based moiré systems for the following reasons. First, monolayer TMD has a large energy gap (1~2eV) between the conduction and valence band, much larger than the interaction strength on moiré scale – therefore the renormalization from the conduction bands can be safely ignored. Second, the lower moiré valence bands could in principle generate considerable renormalization to the topmost moiré band, which is an important problem we plan to study in the future. However, the situation here is different from TBG. The moiré bandwidth in tWSe₂ is large, up to ~100meV for twist angle 5.1 degrees, while the interaction energy is ~35meV. Thus, we expect the renormalization of the band dispersion near half-filling to be small. In summary, although we understand taking the non-interacting dispersion from continuum model as a starting point is an approximation, it is a good one at least in the case of tWSe₂ with large twist angle.

Assuming the remote bands are properly taken care of i.e. we get some dispersion relation for the relevant moiré bands after projecting out the remote bands, there maybe is a separate issue that the people may worry about. Within the band we care, interactions will also lead to renormalization of quasiparticle dispersion. The question is whether one should take the renormalized band or the non-renormalized band structure as the start point to consider various

ordering instabilities. For this, we argue the latter is correct. The former choice will face the problem of double counting the effect of interaction. The modification of quasiparticle dispersion should be a consequence of interaction effect and not treated as a starting point.

Is there any symmetry reason to expect the form of the dispersion in Eq. 1 to survive in the presence of interaction renormalization effects? Or are the authors arguing that there exists some displacement field for which a van Hove singularity emerges regardless of the band structure details?

Indeed, the dispersion in Eq. 1 is the small momentum expansion near K up to third order in k with respect to the C3 rotational symmetry. In another word, Eq. 1 will be the most general form even including interaction renormalizations from remote bands. In this dispersion, the parameter α (or the ratio between coefficient of the k^2 term and that of the k^3 term) is smooth function of the displacement field due to continuity considerations. As α cross zero our system will go through a higher order van hove singularity with a stronger density of state divergence.

4. The authors argue that out-of-plane magnetic field does not act as pair breaking due to Zeeman effect. However, out-of-plane field is also generally expected to couple to orbital degree of freedom leading to a “valley” Zeeman effect which can have a much larger g-factor (See 1908.05110). The authors should comment on whether this effect will lead to pair breaking.

We thank the reviewer for pointing this out. Perhaps our wording here is a bit confusing. In the manuscript, when referring to the Zeeman splitting, we have already taken into account of the valley effect and used the renormalized g-factor. Pair breaking vs non-pair breaking term refers to the form of the term in the basis after particle-hole transformation of one valley (which is the basis where one can make analog to the BCS superconductor). As we listed in table I, the chemical potential term in the original basis maps to precisely to the Zeeman field in BCS language, hence has pair-breaking effect. On the contrary, the Zeeman splitting (including the renormalization) maps to the chemical potential in the BCS basis, hence is non-pair-breaking. Indeed, in our calculation, we observe that the intervalley exciton order is more resilient to the Zeeman splitting than the chemical potential.

To clarify, we add the following sentence on Page 3 under Eq. (5):

“Due to the orbital angular momentum in TMD systems, the holes in the valence band have large renormalized g -factor $\text{cite}\{TMDgfactor\}$. Therefore, The primary effect of a weak magnetic field is Zeeman spin/valley splitting. In contrast to α and μ , the Zeeman coupling maps to the chemical potential in a superconductor and its effect is non-pair-breaking.”

(p.s. 1908.05110 seems to be a reference for algebraic geometry - maybe it’s mistyped. We would like to cite it if the reviewer provides the correct reference. We did cite a reference in our manuscript, Ref [49], for the large renormalized g-factor including the valley effects.)

We thank the reviewer again for these insightful comments. They help us clarify and evaluate the

assumptions in our approach and make our manuscript on a firmer ground. We hope the manuscript is now appropriate for publication in Nature Communication.

Reviewer #3 (Remarks to the Author):

The authors analyze the properties of AA stacked WSe₂ bilayer moiré crystals with a half-filled hole band. The work was motivated by a recent experiment that reported strong insulating behavior over a surprisingly narrow range of external displacement field vanishes. They interpret this finding as being due to the formation of an excitonic density wave state, arguing that this scenario is more likely since the bands are relatively broad, and predict that it will be possible to perform an all-optical experiment that demonstrates spin-superfluidity - which is an expected property of these states. The density wave is stabilized by a nesting condition that is satisfied over a narrow range of displacement fields.

I recommend publication of this manuscript which makes an interesting and testable prediction. The arguments advanced in favor of this prediction are plausible and carefully discussed. I have some suggestions that the author might consider.

We are truly grateful for the wonderful summary of our work and the recommendation of publication from the reviewer. We address the reviewer's comments in the following point by point.

i) The second last paper in the abstract is awkward and needs to be rewritten.

We thank the referee for the valuable suggestion. Indeed the original sentence is confusing. We have revised the sentence to the following hoping it sets things clear. The revised sentence is "Our theory explains the remarkable sensitivity of the insulating gap to the vertical electric field. In contrast, our theory shows that the insulating gap is reduced mildly by a perpendicular magnetic field, with quadratic dependence at low field. These physics can be understood in terms of pair-breaking versus non-pair-breaking effects in a BCS analog of the system."

ii) The authors refer to the state they propose as an excitonic density-wave. It may be that this terminology is perfectly standard - but it seems to me that it requires a few comments. I think that they are saying that their state is a spin and charge density wave state of the type that can be viewed as an exciton condensate. Is there at least a reference that could be cited where the meaning of this state name is carefully defined? In any event, because it is a density wave, collective transport is pinned. The authors do account for relaxation between valleys when they consider transport, but the flip side of this is coupling between the valleys in the ground state. I believe that there are strong arguments related to momentum conservation that these effects are weak - but perhaps this should be discussed explicitly.

We thank the reviewer's excellent suggestion. In our definition, the exciton density wave refers to an exciton condensate that occurs at finite momentum. This phenomenon indeed existed in the literature before. The earliest one we can find is Phys. Rev. Lett. 67, 895, which discussed the possible appearance of exciton density wave state in double-quantum-well in strong magnetic

field (we now add this to the reference list as Ref[49]). The order parameter in our system is very similar to the one in this earlier work by identifying the valley degree of freedom as the layer index.

The TMD materials that can exhibit moiré physics are of very high quality. There could be some distortion of the moiré superlattices but the disorder at atomic scales are small. The order parameter is the pairing between the particles and holes from different valleys which have large momentum distance. Therefore, the exciton density wave will have oscillations at atomic scales. Since the disorder on such scale is small, we expect the pinning effect is weak. This is equivalent to the statement that the valley conservation symmetry is a good approximate symmetry in the system.

REVIEWERS' COMMENTS

Reviewer #2 (Remarks to the Author):

The authors have addressed all my concerns satisfactorily. I recommend for publication.

Reviewer #3 (Remarks to the Author):

The authors have made a thorough and thoughtful response to all referee comments. They have proposed a plausible explanation of the a recent set of detailed observations, and explained how the proposed explanation can be tested experimentally. The revised MS is improved and will be of interest to many readers. I recommend that it be accepted for publication.

Response to Reviewers' comments (NCOMMS-20-23082B)

Referees' comments:

Reviewer #2 (Remarks to the Author):

The authors have addressed all my concerns satisfactorily. I recommend for publication.

Reviewer #3 (Remarks to the Author):

The authors have made a thorough and thoughtful response to all referee comments. They have proposed a plausible explanation of a recent set of detailed observations, and explained how the proposed explanation can be tested experimentally. The revised MS is improved and will be of interest to many readers. I recommend that it be accepted for publication.

We sincerely thank all the reviewers for taking their time to carefully review our work again. We are pleased to see that our previous reply is satisfactory to all referees.

We add two sentences to clarify the difference of the valley-polarized state and intervalley exciton state as well as the experimental signature to distinguish them (in the paragraph before section II C.):

“Experimentally, these two states can be very easily distinguished by their response to an out-of-plane magnetic field. The valley-polarized state features spontaneous magnetization, hysteretic behavior and (quantized) anomalous hall effect, and it is stabilized by the field. On the contrary, the intervalley exciton insulator is weakened under the field and eventually transitions into the valley-polarized insulator above a critical field.”

We have also improved the image quality. We add a short summary for each figure in the caption as required by the policies of nature communication.

Data and Code availability, author contribution and competing interests are also updated at the end of the draft.